# Smooth markets: A basic mechanism for organizing gradient-based learners

**David Balduzzi**[1]**, Wojciech M Czarnecki**[1]**, Thomas W Anthony**[1]**, Ian M Gemp**[1]**,
Edward Hughes**[1]**, Joel Z Leibo**[1]**, Georgios Piliouras**[2]**, Thore Graepel**[1]
[1]DeepMind <google.com>; [2]Singapore University of Technology and Design <sutd.edu.sg>
{dbalduzzi,lejlot,twa,imgemp,edwardhughes,jzl,georgios,thore}@*

## ABSTRACT

With the success of modern machine learning, it is becoming increasingly important to understand and control how learning algorithms interact. Unfortunately, negative results from game theory show there is little hope of understanding or controlling general $n$-player games. We therefore introduce smooth markets (SM-games), a class of $n$-player games with pairwise zero sum interactions. SM-games codify a common design pattern in machine learning that includes (some) GANs, adversarial training, and other recent algorithms. We show that SM-games are amenable to analysis and optimization using first-order methods.

*"I began to see **legibility** as a central problem in modern statecraft. The premodern state was, in many respects, partially blind [. . .] It lacked anything like a detailed 'map' of its terrain and its people. It lacked, for the most part, a measure, a metric that would allow it to 'translate' what it knew into a common standard necessary for a synoptic view. As a result, its interventions were often crude and self-defeating."*
– from **Seeing like a State** by Scott (1999)

## 1 INTRODUCTION

As artificial agents proliferate, it is increasingly important to analyze, predict and control their collective behavior (Parkes and Wellman, 2015; Rahwan et al., 2019). Unfortunately, despite almost a century of intense research since von Neumann (1928), game theory provides little guidance outside a few special cases such as two-player zero-sum, auctions, and potential games (Monderer and Shapley, 1996; Nisan et al., 2007; Vickrey, 1961; von Neumann and Morgenstern, 1944). Nash equilibria provide a general solution concept, but are intractable in almost all cases for many different reasons (Babichenko, 2016; Daskalakis et al., 2009; Hart and Mas-Colell, 2003). These and other negative results (Palaiopanos et al., 2017) suggest that understanding and controlling societies of artificial agents is near hopeless. Nevertheless, human societies – of billions of agents – manage to organize themselves reasonably well and mostly progress with time, suggesting game theory is missing some fundamental organizing principles.

In this paper, we investigate how markets structure the behavior of agents. Market mechanisms have been studied extensively (Nisan et al., 2007). However, prior work has restricted to concrete examples, such as auctions and prediction markets, and strong assumptions, such as convexity. Our approach is more abstract and more directly suited to modern machine learning where the building blocks are neural nets. Markets, for us, encompass discriminators and generators trading errors in GANs (Goodfellow et al., 2014) and agents trading wins and losses in StarCraft (Vinyals et al., 2019).

### 1.1 OVERVIEW

The paper introduces a class of games where **optimization and aggregation make sense**. The phrase requires unpacking. "Optimization" means gradient-based methods. Gradient descent (and friends) are the workhorse of modern machine learning. Even when gradients are not available, gradient *estimates* underpin many reinforcement learning and evolutionary algorithms. "Aggregation" means weighted sums. Sums and averages are the workhorses for analyzing ensembles and populations

across many fields. "Makes sense" means we can draw conclusions about the gradient-based dynamics of the collective by summing over properties of its members.

As motivation, we present some pathologies that arise in even the simplest smooth games. Examples in section 2 show that coupling strongly concave profit functions to form a game can lead to uncontrolled behavior, such as spiraling to infinity and excessive sensitivity to learning rates. Hence, one of our goals is to understand how to 'glue together agents' such that their collective behavior is predictable.

Section 3 introduces a class of games where simultaneous gradient ascent behaves well and is amenable to analysis. In a **smooth market (SM-game)**, each player's profit is composed of a personal objective and pairwise zero-sum interactions with other players. Zero-sum interactions are analogous to monetary exchange (my expenditure is your revenue), double-entry bookkeeping (credits balance debits), and conservation of energy (actions cause equal and opposite reactions). *SM-games explicitly account for externalities*. Remarkably, building this simple bookkeeping mechanism into games has strong implications for the dynamics of gradient-based learners. SM-games generalize adversarial games (Cai et al., 2016) and codify a common **design pattern** in machine learning, see section 3.1.

Section 4 studies SM-games from two points of view. Firstly, from that of a rational, profit-maximizing agent that makes decisions based on first-order profit **forecasts**. Secondly, from that of the game as a whole. SM-games are not potential games, so the game does not optimize any single function. A collective of profit-maximizing agents is *not* rational because they do not optimize a shared objective (Drexler, 2019). We therefore introduce the notion of **legibility**, which quantifies how the dynamics of the collective relate to that of individual agents.

Finally, section 5 applies legibility to prove some basic theorems on the dynamics of SM-games under gradient-ascent. We show that **(i)** Nash equilibria are stable; **(ii)** that if profits are strictly concave then gradient ascent converges to a Nash equilibrium for all learning rates; and **(iii)** the dynamics are bounded under reasonable assumptions.

The results are important for two reasons. Firstly, we identify a class of games whose dynamics are, at least in some respects, amenable to analysis and control. The kinds of pathologies described in section 2 cannot arise in SM-games. Secondly, we identify the specific quantities, forecasts, that are useful to track at the level of individual firms and can be meaningfully aggregated to draw conclusions about their global dynamics. It follows that forecasts should be a useful lever for mechanism design.

## 1.2 RELATED WORK

A wide variety of machine learning markets and agent-based economies have been proposed and studied: Abernethy and Frongillo (2011); Balduzzi (2014); Barto et al. (1983); Baum (1999); Hu and Storkey (2014); Kakade et al. (2003; 2005); Kearns et al. (2001); Kwee et al. (2001); Lay and Barbu (2010); Minsky (1986); Selfridge (1958); Storkey (2011); Storkey et al. (2012); Sutton et al. (2011); Wellman and Wurman (1998). The goal of this paper is different. Rather than propose another market mechanism, we abstract an existing design pattern and elucidate some of its consequences for interacting agents.

Our approach draws on work studying convergence in generative adversarial networks (Balduzzi et al., 2018; Gemp and Mahadevan, 2018; Gidel et al., 2019; Mescheder, 2018; Mescheder et al., 2017), related minimax problems (Abernethy et al., 2019; Bailey and Piliouras, 2018), and monotone games (Gemp and Mahadevan, 2017; Nemirovski et al., 2010; Tatarenko and Kamgarpour, 2019).

## 1.3 CAVEAT

We consider dynamics in continuous time $\frac{d\mathbf{w}}{dt} = \boldsymbol{\xi}(\mathbf{w})$ in this paper. Discrete dynamics, $\mathbf{w}_{t+1} \leftarrow \mathbf{w}_t + \boldsymbol{\xi}(\mathbf{w})$ require a more delicate analysis, e.g. Bailey et al. (2019). In particular, we do not claim that optimizing GANs and SM-games is easy in discrete time. Rather, our analyis shows that it is relatively easy in continuous time, and therefore possible in discrete time, with some additional effort. The contrast is with smooth games in general, where gradient-based methods have essentially no hope of finding local Nash equilibria even in continuous time.

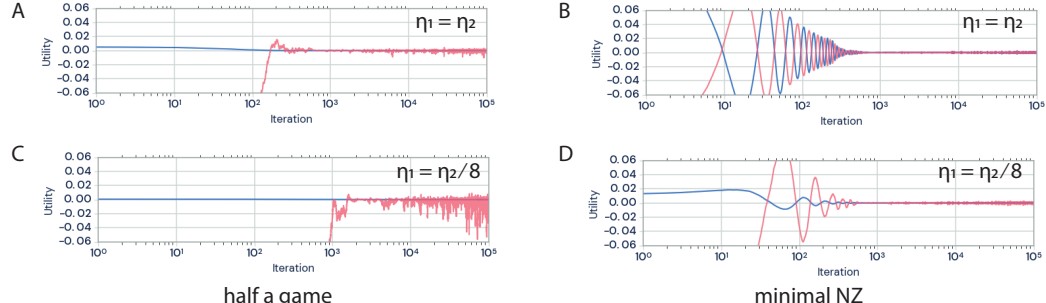

Figure 1: **Effect of learning rates in two games.** Note: $x$-axis is log-scale. *Left:* "half a game", e.g. 2. *Right:* minimal SM-game, e.g. 3. *Top:* Both players have same learning rate. *Bottom:* Second player has $\frac{1}{8}$ learning rate of first (which is same as for top). Reducing the learning rate of the second player destabilizes the dynamics in "half a game", whereas the SM-game is essentially unaffected.

## 1.4 NOTATION

Vectors are column-vectors. The notations $\mathbf{S} \succ \mathbf{0}$ and $\mathbf{v} \succ \mathbf{0}$ refer to a positive-definite matrix and vector with all entries positive respectively. Rather than losses, we work with profits. Proofs are in the appendix. We use economic terminology (firms, profits, forecasts, and sentiment) even though the examples of SM-games, such as GANs and adversarial training, are taken from mainstream machine learning. We hope the economic terminology provides an invigorating change of perspective. The underlying mathematics is no more than first and second-order derivatives.

| | | |
|---|---|---|
| profit of firm $i$ | $\pi_i(\mathbf{w})$ | (1) |
| gradient of profit | $\boldsymbol{\xi}_i(\mathbf{w}) := \nabla_{\mathbf{w}_i}\pi_i(\mathbf{w})$ | |
| profit forecast | $\mathfrak{f}_{\mathbf{v}_i}(\mathbf{w}) := \mathbf{v}_i^\mathsf{T} \cdot \boldsymbol{\xi}_i(\mathbf{w})$ | (2) |
| aggregate forecast | $\sum_i \mathfrak{f}_{\mathbf{v}_i}(\mathbf{w}_i)$ | (3) |
| sentiment of firm $i$ | $\mathbf{v}_i^\mathsf{T} \cdot \nabla_{\mathbf{w}_i}\mathfrak{f}_{\mathbf{v}_i}(\mathbf{w})$ | (4) |

## 2 SMOOTH GAMES

Smooth games model interacting agents with differentiable objectives. They are the kind of games that are played by neural nets. In practice, the differentiability assumption can be relaxed by replacing gradients with gradient estimates.

**Definition 1.** *A **smooth game** (Letcher et al., 2019) consists in $n$ players $[n] = \{1, \dots, n\}$, equipped with twice continuously differentiable profit functions $\{\pi_i : \mathbb{R}^d \to \mathbb{R}\}_{i=1}^n$. The parameters are $\mathbf{w} = (\mathbf{w}_1, \dots, \mathbf{w}_n) \in \mathbb{R}^d$ with $\mathbf{w}_i \in \mathbb{R}^{d_i}$ where $\sum_{i=1}^n d_i = d$. Player $i$ controls the parameters $\mathbf{w}_i$.*

*If players update their actions via **simultaneous gradient ascent**, then a smooth game yields a **dynamical system** specified by the differential equation $\frac{d\mathbf{w}}{dt} = \boldsymbol{\xi}(\mathbf{w})$ for*

$$\boldsymbol{\xi}(\mathbf{w}) := \left(\boldsymbol{\xi}_1(\mathbf{w}), \dots, \boldsymbol{\xi}_n(\mathbf{w})\right)$$

*where $\boldsymbol{\xi}_i(\mathbf{w}) := \nabla_{\mathbf{w}_i}\pi_i(\mathbf{w})$ is a $d_i$-vector. The **Jacobian** of a game is the $(d \times d)$-matrix of second-derivatives $\mathbf{J}(\mathbf{w}) := \left(\frac{\partial \xi_\alpha(\mathbf{w})}{\partial w_\beta}\right)_{\alpha,\beta=1}^d$. The setup can be recast in terms of minimizing losses by substituting $\ell_i := -\pi_i$ for all $i$.*

Smooth games are too general to be tractable since they encompass all dynamical systems.

**Lemma 1.** *Every continuous dynamical system on $\mathbb{R}^d$, for any $d$, arises as simultaneous gradient ascent on the profit functions of a smooth game.*

The next two sections illustrate some problems that arise in simple smooth games.

**Definition 2.** *We recall some solution concepts from dynamical systems and game theory:*

- *A **stable fixed point**[1] $\mathbf{w}^*$ satisfies $\boldsymbol{\xi}(\mathbf{w}^*) = 0$ and $\mathbf{v}^\intercal \cdot \mathbf{J}(\mathbf{w}^*) \cdot \mathbf{v} < 0$ for all vectors $\mathbf{v} \neq \mathbf{0}$.*

- *A **local Nash equilibrium** $\mathbf{w}^*$ has neighborhoods $U_i$ of $\mathbf{w}_i^*$ for all $i$, such that $\pi_i(\mathbf{w}_i, \mathbf{w}_{-i}^*) < \pi_i(\mathbf{w}_i^*, \mathbf{w}_{-i}^*)$ all $\mathbf{w}_i \in U_i \setminus \{\mathbf{w}_i^*\}$.*

- *A **classical Nash equilibrium** $\mathbf{w}^*$ satisfies $\pi_i(\mathbf{w}_i, \mathbf{w}_{-i}^*) \leq \pi_i(\mathbf{w}_i^*, \mathbf{w}_{-i}^*)$ for all $\mathbf{w}_i$ and all players $i$.*

Example 1 below shows that stable fixed points and local Nash equilibria do not necessarily coincide. The notion of classical Nash equilibrium is ill-suited to nonconcave settings.

Intuitively, a fixed point is stable if all trajectories sufficiently nearby flow into it. A joint strategy is a local Nash if each player is harmed if it makes a small unilateral deviation. Local Nash differs from the classic definition in two ways. It is weaker, because it only allows *small* unilateral deviations. This is necessary since players are neural networks and profits are not usually concave. It is also stronger, because unilateral deviations decrease (rather than *not increase*) profits.

## 2.1 PROBLEMS WITH POTENTIAL GAMES

A game is a potential game if $\boldsymbol{\xi} = \nabla\phi$ for some function $\phi$, see Balduzzi et al. (2018) for details.

**Example 1** (potential game). *Fix a small $\epsilon > 0$. Consider the two-player games with profit functions*

$$\pi_1(\mathbf{w}) = w_1 w_2 - \frac{\epsilon}{2} w_1^2 \text{ and } \pi_2(\mathbf{w}) = w_1 w_2 - \frac{\epsilon}{2} w_2^2.$$

*The game has a unique local Nash equilibrium at $\mathbf{w} = (0, 0)$ with $\pi_1(0, 0) = 0 = \pi_2(0, 0)$.*

The game is chosen to be as nice as possible: $\pi_1$ and $\pi_2$ are strongly concave functions of $w_1$ and $w_2$ respectively. The game is a **potential game** since $\boldsymbol{\xi} = (w_2 - \epsilon w_1, w_1 - \epsilon w_2) = \nabla\phi$ for $\phi(\mathbf{w}) = w_1 w_2 - \frac{\epsilon}{2}(w_1^2 + w_2^2)$. Nevertheless, the game exhibits three related problems.

Firstly, *the Nash equilibrium is unstable.* Players at the Nash equilibrium can increase their profits via the joint update $\mathbf{w} \leftarrow (0, 0) + \eta \cdot (1, 1)$, so $\pi_1(\mathbf{w}) = \eta(1 - \frac{\epsilon}{2}) = \pi_2(\mathbf{w}) > 0$. The existence of a Nash equilibrium where players can improve their payoffs by *coordinated action* suggests the incentives are not well-designed.

Secondly, *the dynamics can diverge to infinity.* Starting at $\mathbf{w}^{(1)} = (1, 1)$ and applying simultaneously gradient ascent causes the norm of vector $\|\mathbf{w}^{(t)}\|_2$ to increase without limit as $t \to \infty$ – and at an accelerating rate – due to a positive feedback loop between the players' parameters and profits. Finally, *players impose externalities on each other.* The decisions of the first player affect the profits of the second, and vice versa. Obviously players must interact for a game to be interesting. However, positive feedback loops arise because the interactions are not properly accounted for.

In short, simultaneous gradient ascent does not converge to the Nash – and can diverge to infinity. It is open to debate whether the fault lies with gradients, the concept of Nash, or the game structure. In this paper, we take gradients and Nash equilibria as given and seek to design better games.

## 2.2 PROBLEMS WITH LEARNING RATES

Gradient-based optimizers rarely follow the actual gradient. For example RMSProp and Adam use adaptive, parameter-dependent learning rates. This is not a problem when optimizing a function. Suppose $f(\mathbf{w})$ is optimized with reweighted gradient $(\nabla f)_{\boldsymbol{\eta}} := (\eta_1 \nabla_1 f, \ldots, \eta_n \nabla_n f)$ where $\boldsymbol{\eta} \succ \mathbf{0}$ is a vector of learning rates. Even though $(\nabla f)_{\boldsymbol{\eta}}$ is not necessarily the gradient of *any* function, it behaves *like* $\nabla f$ because they have positive inner product when $\nabla f \neq \mathbf{0}$:

$$(\nabla f)_{\boldsymbol{\eta}}^\intercal \cdot \nabla f = \sum_i \eta_i \cdot (\nabla_i f)^2 > 0, \text{ since } \eta_i > 0 \text{ for all } i.$$

Parameter-dependent learning rates thus behave well in *potential* games where the dynamics derive from an implicit potential function $\boldsymbol{\xi}(\mathbf{w}) = \nabla\phi(\mathbf{w})$. Severe problems can arise in general games.

---

[1]Berard et al. (2019) use a different notion of stable fixed point that requires $\mathbf{J}$ has positive eigenvalues.

**Example 2** ("half a game"). *Consider the following game, where the $w_2$-player is indifferent to $w_1$:*

$$\pi_1(\mathbf{w}) = w_1 w_2 - \frac{\epsilon}{2} w_1^2 \text{ and } \pi_2(\mathbf{w}) = -\frac{\epsilon}{2} w_2^2.$$

The dynamics are clear by inspection: the $w_2$-player converges to $w_2 = 0$, and then the $w_1$-player does the same. It is hard to imagine that anything could go wrong. In contrast, behavior in the next example should be worse because convergence is slowed down by cycling around the Nash:

**Example 3** (minimal SM-game). *A simple SM-game, see definition 3, is*

$$\pi_1(\mathbf{w}) = w_1 w_2 - \frac{\epsilon}{2} w_1^2 \text{ and } \pi_2(\mathbf{w}) = -w_1 w_2 - \frac{\epsilon}{2} w_2^2.$$

Figure 1 shows the dynamics of the games, in discrete time, with small learning rates and small gradient noise. In the top panel, both players have the same learning rate. Both games converge. Example 2 converges faster – as expected – without cycling around the Nash.

In the bottom panels, the learning rate of the second player is decreased by a factor of eight. The SM-game's dynamics do not change significantly. In contrast, the dynamics of example 2 become unstable: although player 1 is attracted to the Nash, it is extremely sensitive to noise and does not stay there for long. One goal of the paper is to explain why SM-games are more robust, in general, to differences in relative learning rates.

### 2.3 STOP_GRADIENT AND LEARNING RATES

Tools for automatic differentiation (AD) such as TensorFlow and PyTorch include `stop_gradient` operators that stop gradients from being computed. For example, let $f(\mathbf{w}) = w_1 \cdot$ `stop_gradient`$(w_2) - \frac{\epsilon}{2}(w_1^2 + w_2^2)$. The use of `stop_gradient` means $f$ is not strictly speaking a function and so we use $\nabla_{\text{AD}}$ to refer to its gradient under automatic differentiation. Then

$$\nabla_{\text{AD}} f(\mathbf{w}) = (w_2 - \epsilon w_1, -\epsilon w_2)$$

which is the simultaneous gradient from example 2. Any smooth vector field is the gradient of a function augmented with `stop_gradient` operators, see appendix D. `Stop_gradient` is often used in complex neural architectures (for example when one neural network is fed into another leading to multiplicative interactions), and is thought to be mostly harmless. Section 2.2 shows that `stop_gradients` can interact in unexpected ways with parameter-dependent learning rates.

### 2.4 SUMMARY

It is natural to expect individually well-behaved agents to also behave well collectively. Unfortunately, this basic requirement fails in even the simplest examples.

Maximizing a strongly concave function is well-behaved: there is a unique, *finite* global maximum. However, example 1 shows that coupling concave functions can cause simultaneous gradient ascent to diverge to infinity. The dynamics of the game **differs in kind** from the dynamics of the players in isolation. Example 2 shows that *reducing* the learning rate of a well-behaved (strongly concave) player in a simple game destabilizes the dynamics. How collectives behave is sensitive not only to profits, but also to relative learning rates. Off-the-shelf optimizers such as Adam (Kingma and Ba, 2015) modify learning rates under the hood, which may destabilize some games.

## 3 SMOOTH MARKETS (SM-GAMES)

Let us restrict to more structured games. Take an accountant's view of the world, where the only thing we track is the flow of money. Interactions are pairwise. Money is neither created nor destroyed, so interactions are zero-sum. If we model the interactions between players by differentiable functions $g_{ij}(\mathbf{w}_i, \mathbf{w}_j)$ that depend on their respective strategies then we have an SM-game. All interactions are explicitly tracked. There are no externalities off the books. Positive interactions, $g_{ij} > 0$, are revenue, negative are costs, and the difference is profit. The model prescribes that all firms are profit maximizers. More formally:

**Definition 3** (SM-game). *A **smooth market** is a smooth game where interactions between players are pairwise zero-sum. The profits have the form*

$$\pi_i(\mathbf{w}) = f_i(\mathbf{w}_i) + \sum_{j \neq i} g_{ij}(\mathbf{w}_i, \mathbf{w}_j) \tag{1}$$

*where $g_{ij}(\mathbf{w}_i, \mathbf{w}_j) + g_{ji}(\mathbf{w}_j, \mathbf{w}_i) \equiv 0$ for all $i, j$.*

The functions $f_i$ can act as regularizers. Alternatively, they can be interpreted as natural resources or dummy players that react too slowly to model as players. Dummy players provide firms with easy (non-adversarial) sources of revenue.

Humans, unlike firms, are not profit-maximizers; humans typically buy goods because they value them more than the money they spend on them. Appendix C briefly discusses extending the model.

### 3.1 EXAMPLES OF SM-GAMES

SM-games codify a common design pattern:

1. *Optimizing a function.* A near-trivial case is where there is a single player with profit $\pi_1(\mathbf{w}) = f_1(\mathbf{w})$.

2. *Generative adversarial networks* and related architectures like CycleGANs are zero or near zero sum (Goodfellow et al., 2014; Wu et al., 2019; Zhu et al., 2017).

3. *Zero-sum polymatrix games* are SM-games where $f_i(\mathbf{w}_i) \equiv 0$ and $g_{ij}(\mathbf{w}_i, \mathbf{w}_j) = \mathbf{w}_i^\mathsf{T} \mathbf{A}_{ij} \mathbf{w}_j$ for some matrices $\mathbf{A}_{ij}$. Weights are constrained to probability simplices. The games have nice properties including: Nash equilibria are computed via a linear program and correlated equilibria marginalize onto Nash equilibria (Cai et al., 2016).

4. *Intrinsic curiosity modules* use games to drive exploration. One module is rewarded for predicting the environment and an adversary is rewarded for choosing actions whose outcomes are *not* predicted by the first module (Pathak et al., 2017). The modules share some weights, so the setup is nearly, but not exactly, an SM-game.

5. *Adversarial training* is concerned with the minmax problem (Kurakin et al., 2017; Madry et al., 2018)

$$\min_{\mathbf{w} \in W} \sum_i \left[ \max_{\boldsymbol{\delta}_i \in B_\epsilon} \ell\Big(f_\mathbf{w}(\mathbf{x}_i + \boldsymbol{\delta}_i), y_i\Big) \right].$$

Setting $g_{0i}(\mathbf{w}_0, \boldsymbol{\delta}_i) = \ell\big(f_{\mathbf{w}_0}(\mathbf{x}_i + \boldsymbol{\delta}_i), y_i\big)$ obtains a star-shaped SM-game with the neural net (player 0) at the center and $n$ adversaries – one per datapoint $(\mathbf{x}_i, y_i)$ – on the arms.

6. *Task-suites* where a population of agents are trained on a population of tasks, form a bipartite graph. If the tasks are *parametrized* and *adversarially* rewarded based on their difficulty for agents, then the setup is an SM-game.

7. *Homogeneous games* arise when all the coupling functions are equal up to sign (recall $g_{ij} = -g_{ji}$). An example is population self-play (Silver et al., 2016; Vinyals et al., 2019) which lives on a graph where $g_{ij}(\mathbf{w}_i, \mathbf{w}_j) := P(\mathbf{w}_i \text{ beats } \mathbf{w}_j) - \frac{1}{2}$ comes from the probability that policy $\mathbf{w}_i$ beats $\mathbf{w}_j$.

Monetary exchanges in SM-games are quite general. The error signals traded between generators and discriminators and the wins and losses traded between agents in StarCraft are two very different special cases.

## 4 FROM MICRO TO MACRO

How to analyze the behavior of the market as a whole? Adam Smith claimed that profit-maximizing leads firms to promote the interests of society, as if by an invisible hand (Smith, 1776). More formally, we can ask: Is there a measure that firms collectively increase or decrease? It is easy to see that firms do not collectively maximize aggregate profit (AP) or aggregate revenue (AR):

$$\text{AP}(\mathbf{w}) := \sum_i \pi_i(\mathbf{w}) = \sum_i f_i(\mathbf{w}_i) \qquad \text{AR}(\mathbf{w}) := \sum_{i,j} \max\Big(g_{ij}(\mathbf{w}_i, \mathbf{w}_j), 0\Big).$$

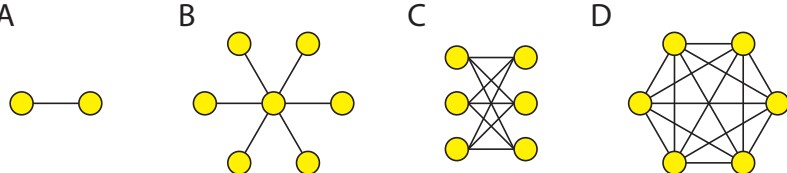

Figure 2: **SM-game graph topologies.** *A:* two-player (e.g. GANs). *B:* star-shaped (e.g. adversarial training). *C:* bipartite (e.g. task-suites). *D:* all-to-all.

Maximizing aggregate profit would require firms to ignore interactions with other firms. Maximizing aggregate revenue would require firms to ignore costs. In short, SM-games are *not* potential games; there is no function that they optimize in general. However, it turns out the dynamics of SM-games aggregates the dynamics of individual firms, in a sense made precise in section 4.3.

### 4.1 RATIONALITY: SEEING LIKE A FIRM

Give an objective function to an agent. The **agent is rational**, relative to the objective, if it chooses actions because it forecasts they will lead to better outcomes as measured by the objective. In SM-games, agents are firms, the objective is profit, and forecasts are computed using gradients.

Firms aim to increase their profit. Applying the first-order Taylor approximation obtains

$$\underbrace{\pi_i(\mathbf{w} + \mathbf{v}_i) - \pi_i(\mathbf{w})}_{\text{change in profit}} = \underbrace{\mathbf{v}_i^\mathsf{T} \boldsymbol{\xi}_i(\mathbf{w})}_{\text{profit forecast}} + \{h.o.t.\}, \tag{2}$$

where $\{h.o.t.\}$ refers to higher-order terms. Firm $i$'s **forecast** of how profits will change if it modifies production by $\mathbf{v}_i$ is $\mathfrak{f}_{\mathbf{v}_i}(\mathbf{w}) := \mathbf{v}_i^\mathsf{T} \boldsymbol{\xi}_i(\mathbf{w})$. The Taylor expansion implies that $\mathfrak{f}_{\mathbf{v}_i}(\mathbf{w}) \approx \pi_i(\mathbf{w} + \mathbf{v}_i) - \pi_i(\mathbf{w})$ for small updates $\mathbf{v}_i$. Forecasts encode how individual firms expect profits to change *ceteris paribus*[2].

### 4.2 PROFIT CHANGES DO NOT ADD UP

How does profit maximizing by individual firms look from the point of view of the market as a whole? Summing over all firms obtains

$$\underbrace{\sum_i \left[ \pi_i(\mathbf{w} + \mathbf{v}_i) - \pi_i(\mathbf{w}) \right]}_{\text{aggregate change in profit}} = \underbrace{\sum_i \mathfrak{f}_{\mathbf{v}_i}(\mathbf{w})}_{\text{aggregate forecast}} + \{h.o.t.\} \tag{3}$$

where $\mathfrak{f}_{\mathbf{v}}(\mathbf{w}) = \sum_i \mathfrak{f}_{\mathbf{v}_i}(\mathbf{w})$ is the **aggregate forecast**. Unfortunately, the left-hand side of Eq. (3) is incoherent. It sums the *changes in profit that would be experienced by firms updating their production in isolation*. However, firms change their production simultaneously. Updates are **not** *ceteris paribus* and so profit is not a meaningful macroeconomic concept. The following minimal example illustrates the problem:

**Example 4.** *Suppose* $\pi_1(\mathbf{w}) = w_1 w_2$ *and* $\pi_2(\mathbf{w}) = -w_1 w_2$. *Fix* $\mathbf{w} = (w_1, w_2)$ *and let* $\mathbf{v} = (w_2, -w_1)$. *The sum of the changes in profit expected by the firms, reasoning in isolation, is*

$$\left[ \pi_1(\mathbf{w} + \mathbf{v}_1) - \pi_1(\mathbf{w}) + \pi_2(\mathbf{w} + \mathbf{v}_2) - \pi_2(\mathbf{w}) \right] = w_2^2 + w_1^2 > 0$$

*whereas the actual change in aggregate profit is zero because* $\pi_1(\mathbf{x}) + \pi_2(\mathbf{x}) = 0$ *for any* $\mathbf{x}$.

Tracking aggregate profits is therefore not useful. The next section shows forecasts are better behaved.

### 4.3 LEGIBILITY: SEEING LIKE AN ECONOMY

Give a target function to every agent in a collective. The **collective is legible**, relative to the targets, if it increases or decreases the aggregate target according to whether its members forecast, on aggregate,

---

[2]All else being equal – i.e. without taking into account updates by other firms.

they will increase or decrease their targets. We show that SM-games are legible. The targets are profit forecasts (note: *not* profits).

Let us consider how forecasts change. Define the **sentiment** as the directional derivative of the forecast $D_{\mathbf{v}_i}\mathfrak{f}_{\mathbf{v}_i}(\mathbf{w}) = \mathbf{v}_i^{\mathsf{T}}\nabla\mathfrak{f}_{\mathbf{v}_i}(\mathbf{w})$. The first-order Taylor expansion of the forecast shows that the sentiment is a forecast about the profit forecast:

$$\underbrace{\mathfrak{f}_{\mathbf{v}_i}(\mathbf{w}+\mathbf{v}_i) - \mathfrak{f}_{\mathbf{v}_i}(\mathbf{w})}_{\text{change in profit forecast}} = \underbrace{\mathbf{v}_i^{\mathsf{T}}\nabla\mathfrak{f}_{\mathbf{v}_i}(\mathbf{w})}_{\text{sentiment}} + \{\text{h.o.t.}\}. \tag{4}$$

The perspective of firms can be summarized as:

1. Choose an update direction $\mathbf{v}_i$ that is forecast to increase profit.
2. The firm is then in one of two main regimes:
   a. If sentiment is positive then forecasts increase as the firm modifies its production – forecasts become more optimistic. The firm experiences **increasing returns-to-scale**.
   b. If sentiment is negative then forecasts decrease as the firm modifies its production – forecasts become more pessimistic. The firm experiences **diminishing returns-to-scale**.

Our main result is that sentiment is additive, which means that forecasts are legible:

**Proposition 2** (forecasts are legible in SM-games). *Sentiment is additive*

$$D_{\mathbf{v}}\mathfrak{f}_{\mathbf{v}}(\mathbf{w}) = \sum_i D_{\mathbf{v}_i}\mathfrak{f}_{\mathbf{v}_i}(\mathbf{w}).$$

*Thus, the aggregrate profit forecast $\mathfrak{f}_{\mathbf{v}}$ increases or decreases according to whether individual forecasts $\mathfrak{f}_{\mathbf{v}_i}$ are expected to increase or decrease in aggregate.*

Section 5.1 works through an example that is *not* legible.

## 5 Dynamics of Smooth Markets

Finally, we study the dynamics of gradient-based learners in SM-games. Suppose firms use gradient ascent. Firm $i$'s updates are, infinitesimally, in the direction $\mathbf{v}_i = \boldsymbol{\xi}_i(\mathbf{w})$ so that $\frac{d\mathbf{w}_i}{dt} = \boldsymbol{\xi}_i(\mathbf{w})$. Since updates are gradients, we can simplify our notation. Define firm $i$'s **forecast** as $\mathfrak{f}_i(\mathbf{w}) := \frac{1}{2}\boldsymbol{\xi}_i^{\mathsf{T}}\nabla_i\pi_i = \frac{1}{2}\|\boldsymbol{\xi}_i(\mathbf{w})\|_2^2$ and its **sentiment**, *ceteris paribus*, as $\boldsymbol{\xi}_i^{\mathsf{T}}\nabla_i\mathfrak{f}_i = \frac{d\mathfrak{f}_i}{dt}(\mathbf{w})$.

We allow firms to choose their learning rates; firms with higher learning rates are more responsive. Define the $\boldsymbol{\eta}$-weighted dynamics $\boldsymbol{\xi}_{\boldsymbol{\eta}}(\mathbf{w}) := (\eta_1\boldsymbol{\xi}_1, \ldots, \eta_n\boldsymbol{\xi}_n)$ and $\boldsymbol{\eta}$-weighted forecast as

$$\mathfrak{f}_{\boldsymbol{\eta}}(\mathbf{w}) := \frac{1}{2}\|\boldsymbol{\xi}_{\boldsymbol{\eta}}(\mathbf{w})\|_2^2 = \frac{1}{2}\sum_i \eta_i^2 \cdot \mathfrak{f}_i(\mathbf{w}).$$

In this setting, proposition 2 implies that

**Proposition 3** (legibility under gradient dynamics). *Fix dynamics $\frac{d\mathbf{w}}{dt} := \boldsymbol{\xi}_{\boldsymbol{\eta}}(\mathbf{w})$. Sentiment decomposes additively:*

$$\frac{d\mathfrak{f}_{\boldsymbol{\eta}}}{dt} = \sum_i \eta_i \cdot \frac{d\mathfrak{f}_i}{dt}$$

Thus, we can read off the aggregate dynamics from the dynamics of forecasts of individual firms.

### 5.1 Example of a failure of legibility

The pairwise zero-sum structure is crucial to legibility. It is instructive to take a closer look at example 1, where the forecasts are *not* legible.

Suppose $\pi_1(\mathbf{w}) = w_1 w_2 - \frac{\epsilon}{2}w_1^2$ and $\pi_2(\mathbf{w}) = w_1 w_2 - \frac{\epsilon}{2}w_2^2$. Then $\boldsymbol{\xi}(\mathbf{w}) = (w_2 - \epsilon w_1, w_1 - \epsilon w_2)$ and the firms' sentiments are $\frac{d\mathfrak{f}_1}{dt} = -\epsilon(w_2 - \epsilon w_1)^2$ and $\frac{d\mathfrak{f}_2}{dt} = -\epsilon(w_1 - \epsilon w_2)^2$ which are always

non-positive. However, the aggregate sentiment is

$$\frac{d\mathfrak{f}}{dt}(\mathbf{w}) = -\epsilon(w_2 - \epsilon w_1)^2 - \epsilon(w_1 - \epsilon w_2)^2 + (1 + \epsilon^2)w_1 w_2 - \epsilon(w_1 + w_2)^2$$

which for small $\epsilon$ is dominated by $w_1 w_2$, and so can be either positive or negative.

When $\mathbf{w} = (1, 1)$ we have $\frac{d\mathfrak{f}}{dt} = 1 - \epsilon(6 - 5\epsilon + 2\epsilon^2) \approx 1 > 0$ and $\frac{d\mathfrak{f}_1}{dt} + \frac{d\mathfrak{f}_2}{dt} = -2\epsilon(1 - \epsilon)^2 < 0$. Each firm expects their forecasts to decrease, and yet the opposite happens due to a positive feedback loop that ultimately causes the dynamics to diverge to infinity.

## 5.2 Stability, Convergence and Boundedness

We provide three fundamental results on the dynamics of smooth markets. Firstly, we show that stability, from dynamical systems, and local Nash equilibrium, from game theory, coincide in SM-games:

**Theorem 4** (stability). *A fixed point in an SM-game is a local Nash equilibrium iff it is stable. Thus, every local Nash equilibrium is contained in an open set that forms its basin of attraction.*

Secondly, we consider convergence. Lyapunov functions are tools for studying convergence. Given dynamical system $\frac{d\mathbf{w}}{dt} = \boldsymbol{\xi}(\mathbf{w})$ with fixed point $\mathbf{w}^*$, recall that $V(\mathbf{w})$ is a **Lyapunov function** if: **(i)** $V(\mathbf{w}^*) = 0$; **(ii)** $V(\mathbf{w}) > 0$ for all $\mathbf{w} \neq \mathbf{w}^*$; and **(iii)** $\frac{dV}{dt}(\mathbf{w}) < 0$ for all $\mathbf{w} \neq \mathbf{w}^*$. If a dynamical system has a Lyapunov function then the dynamics converge to the fixed point. Aggregate forecasts share properties **(i)** and **(ii)** with Lyapunov functions.

 **(i)** **Shared global minima:** $\mathfrak{f}_{\boldsymbol{\eta}}(\mathbf{w}) = 0$ iff $\mathfrak{f}_{\boldsymbol{\eta}'}(\mathbf{w}) = 0$ for all $\boldsymbol{\eta}, \boldsymbol{\eta}' \succ \mathbf{0}$, which occurs iff $\mathbf{w}$ is a stationary point, $\boldsymbol{\xi}_i(\mathbf{w}) = \mathbf{0}$ for all $i$.

 **(ii)** **Positivity:** $\mathfrak{f}_{\boldsymbol{\eta}}(\mathbf{w}) > 0$ for all points that are *not* fixed points, for all $\boldsymbol{\eta} \succ \mathbf{0}$.

We can therefore use forecasts to study convergence and divergence *across all learning rates*:

**Theorem 5.** *In continuous time, for all positive learning rates $\boldsymbol{\eta} \succ \mathbf{0}$,*

 1. ***Convergence:** If $\mathbf{w}^*$ is a stable fixed point ($\mathbf{S} \prec \mathbf{0}$), then there is an open neighborhood $U \ni \mathbf{w}^*$ where $\frac{d\mathfrak{f}_{\boldsymbol{\eta}}}{dt}(\mathbf{w}) < 0$ for all $\mathbf{w} \in U \setminus \{\mathbf{w}^*\}$, so the dynamics converge to $\mathbf{w}^*$ from anywhere in $U$.*

 2. ***Divergence:** If $\mathbf{w}^*$ is an unstable fixed point ($\mathbf{S} \succ \mathbf{0}$), there is an open neighborhood $U \ni \mathbf{w}^*$ such that $\frac{d\mathfrak{f}_{\boldsymbol{\eta}}}{dt}(\mathbf{w}) > 0$ for all $\mathbf{w} \in U \setminus \{\mathbf{w}^*\}$, so the dynamics within $U$ are repelled by $\mathbf{w}^*$.*

The theorem explains why SM-games are robust to relative differences in learning rates – in contrast to the sensitivity exhibited by the game in example 2. If a fixed point is stable, then for any dynamics $\frac{d\mathbf{w}}{dt} = \boldsymbol{\xi}_{\boldsymbol{\eta}}(\mathbf{w})$, there is a corresponding aggregate forecast $\mathfrak{f}_{\boldsymbol{\eta}}(\mathbf{w})$ that can be used to show convergence. The aggregate forecasts provide a family of Lyapunov-like functions.

Finally, we consider the setting where firms experience diminishing returns-to-scale for sufficiently large production vectors. The assumption is realistic for firms in a finite economy since revenues must eventually saturate whilst costs continue to increase with production.

**Theorem 6** (boundedness). *Suppose all firms have negative sentiment for sufficiently large values of $\|\mathbf{w}_i\|$. Then the dynamics are bounded for all $\boldsymbol{\eta} \succ \mathbf{0}$.*

The theorem implies that the kind of positive feedback loops that caused example 1 to diverge to infinity, cannot occur in SM-games.

## 5.3 Legibility and the Landscape

One of our themes is that legibility allows to read off the dynamics of games. We make the claim visually explicit in this section. Let us start with a concrete game.

**Example 5.** *Consider the SM-game with profits*

$$\pi_1(\mathbf{w}) = -\frac{1}{6}|w_1|^3 + \frac{1}{2}w_1^2 - w_1 w_2 \quad and \quad \pi_2(\mathbf{w}) = -\frac{1}{6}|w_2|^3 + \frac{1}{2}w_2^2 + w_1 w_2.$$

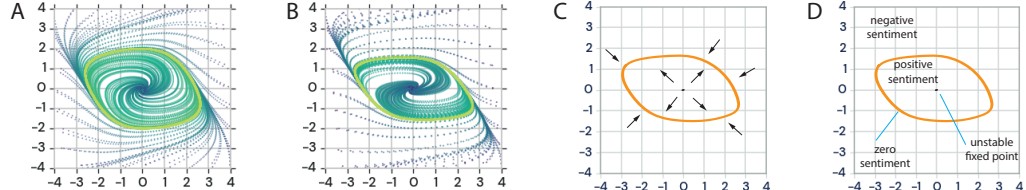

Figure 3: **Legibile dynamics.** *Panels AB:* Dynamics in an SM-game with both positive and negative sentiment, under different learning rates. *Panels CD:* Cartoon maps of the dynamics.

Figure 3AB plots the dynamics of the SM-game in example 5, under two different learning rates for player 1. There is an unstable fixed point at the origin and an ovoidal cycle. Dynamics converge to the cycle from both inside and outside the ovoid. Changing player 1's learning rate, panel B, squashes the ovoid. Panels CD provide a cartoon map of the dynamics. There are two regions, the interior and exterior of the ovoid and the boundary formed by the ovoid itself.

In general, the phase space of any SM-game is carved into regions where sentiment $\frac{d\mathfrak{f}_\eta}{dt}(\mathbf{w})$ is positive and negative, with boundaries where sentiment is zero. The dynamics can be visualized as operating on a landscape where height at each point $\mathbf{w}$ corresponds to the value of the aggregate forecast $\mathfrak{f}_\eta(\mathbf{w})$. The dynamics does not always ascend or always descend the landscape. Rather, sentiment determines whether the dynamics ascends, descends, or remains on a level-set. Since sentiment is additive, $\frac{d\mathfrak{f}_\eta}{dt}(\mathbf{w}) = \sum_i \eta_i \cdot \frac{d\mathfrak{f}}{dt}(\mathbf{w})$, the decision to ascend or descend comes down to a weighted sum of the sentiments of the firms.[3] Changing learning rates changes the emphasis given to different firms' opinions, and thus changes the shapes of the boundaries between regions in a relatively straightforward manner.

SM-games can thus express richer dynamics than potential games (cycles will not occur when performing gradient ascent on a fixed objective), which still admit a relatively simple visual description in terms of a landscape and decisions about which direction to go (upwards or downwards). Computing the landscape for general SM-games, as for neural nets, is intractable.

## 6 DISCUSSION

Machine learning has got a lot of mileage out of treating differentiable modules like plug-and-play lego blocks. This works when the modules optimize a single loss and the gradients chain together seamlessly. Unfortunately, agents with differing objectives are far from plug-and-play. Interacting agents form games, and games are intractable in general. Worse, positive feedback loops can cause individually well-behaved agents to collectively spiral out of control.

It is therefore necessary to find organizing principles – constraints – on how agents interact that ensure their collective behavior is amenable to analysis and control. The pairwise zero-sum condition that underpins SM-games is one such organizing principle, which happens to admit an economic interpretation. Our main result is that SM-games are legible: changes in aggregate forecasts are the sum of how individual firms expect their forecasts to change. It follows that we can translate properties of the individual firms into guarantees on collective convergence, stability and boundedness in SM-games, see theorems 4-6.

Legibility is a local-to-global principle, whereby we can draw qualitative conclusions about the behavior of collectives based on the nature of their individual members. Identifying and exploiting games that embed local-to-global principles will become increasingly important as artificial agents become more common.

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

# APPENDIX

## A  MECHANICS OF SMOOTH MARKETS

This section provides a physics-inspired perspective on smooth markets. Consider a dynamical system with $n$ particles moving according to the differential equations:

$$\frac{d\mathbf{w}_1}{dt} = \boldsymbol{\xi}_1(\mathbf{w}_1, \dots, \mathbf{w}_n)$$
$$\vdots$$
$$\frac{d\mathbf{w}_n}{dt} = \boldsymbol{\xi}_n(\mathbf{w}_1, \dots, \mathbf{w}_n).$$

The kinetic energy of a particle is mass times velocity squared, $mv^2$, or in our case

$$\text{energy of } i^{\text{th}} \text{ particle} = \eta_i^2 \cdot \mathfrak{f}_i(\mathbf{w}) = \eta_i^2 \cdot \|\boldsymbol{\xi}_i\|^2$$

where we interpret the learning rate squared $\eta_i^2$ of particle $i$ as its mass and $\boldsymbol{\xi}_i$ as its velocity. The total energy of the system is the sum over the kinetic energies of the particles:

$$\text{total energy} = \mathfrak{f}_{\boldsymbol{\eta}}(\mathbf{w}) = \sum_i \eta_i^2 \cdot \|\boldsymbol{\xi}_i\|^2.$$

For example, in a Hamiltonian game we have that energy is conserved:

$$\text{total energy} = \|\boldsymbol{\xi}\|^2 = \text{constant} \quad \text{or} \quad \frac{d\mathfrak{f}}{dt}(\mathbf{w}) = 0$$

since $\boldsymbol{\xi}^\mathsf{T} \cdot \nabla \|\boldsymbol{\xi}\|^2 = \boldsymbol{\xi}^\mathsf{T} \cdot \mathbf{A}^\mathsf{T} \boldsymbol{\xi} = 0$, see Balduzzi et al. (2018); Letcher et al. (2019) for details.

Energy is measured in joules ($kg \cdot m \cdot s^{-2}$). The rate of change of energy with respect to time is *power*, measured in joules per second or watts ($kg \cdot m \cdot s^{-3}$). Conservation of energy means that a (closed) Hamiltonian system, in aggregate, generates no power. The existence of an invariant function makes Hamiltonian systems easy to reason about in many ways.

Smooth markets are more general than Hamiltonian games in that total energy is not necessarily conserved. Nevertheless, they are much more constrained than general dynamical systems. Legibility, proposition 3, says that the total power (total rate of energy generation) in smooth markets is the sum of the power (rate of energy generation) of the individual particles:

$$\frac{d\mathfrak{f}}{dt}(\mathbf{w}) = \sum_i \frac{d\mathfrak{f}_i}{dt}(\mathbf{w}).$$

**Example where legibility fails.** Once again, it is instructive to look at a concrete example where legibility fails. Recall the potential game in example 1 with profits

$$\pi_1(\mathbf{w}) = w_1 w_2 - \frac{\epsilon}{2} w_1^2 \quad \text{and} \quad \pi_2(\mathbf{w}) = w_1 w_2 - \frac{\epsilon}{2} w_2^2.$$

and sentiments

$$\frac{d\mathfrak{f}_1}{dt}(\mathbf{w}) = -\epsilon(w_2 - \epsilon w_1)^2 \quad \text{and} \quad \frac{d\mathfrak{f}_2}{dt}(\mathbf{w}) = -\epsilon(w_1 - \epsilon w_2)^2.$$

Physically, the negative sentiments $\frac{d\mathfrak{f}_1}{dt} < 0$ and $\frac{d\mathfrak{f}_2}{dt} < 0$ mean that that each "particle" in the system, considered in isolation, is always dissipating energy. Nevertheless as shown in section 5.1 the system as a whole has

$$\frac{d\mathfrak{f}}{dt}(\mathbf{w}) = -\epsilon(w_2 - \epsilon w_1)^2 - \epsilon(w_1 - \epsilon w_2)^2 + (1 + \epsilon^2)w_1 w_2 - \epsilon(w_1 + w_2)^2$$

which is positive for some values of $\mathbf{w}$. Thus, the system as a whole can generate energy through interaction effects between the (dissipative) particles.

# B  PROOFS

**Proof of lemma 1.**

**Lemma 1.** *Every continuous dynamical system on $\mathbb{R}^d$, for any d, arises as simultaneous gradient ascent on the profit functions of a smooth game.*

*Proof.* Specifically, we mean that every dynamical system of the form $\frac{d\mathbf{w}}{dt} = \boldsymbol{\xi}(\mathbf{w})$ arises as simultaneous gradient ascent on the profits of a smooth game.

Given continuous vector field $\boldsymbol{\xi}$ on $\mathbb{R}^d$, we need to construct a smooth game with dynamics given by $\boldsymbol{\xi}$. To that end, consider a $d$-player game where player $i$ controls coordinate $w_i$. Set the profit of player $i$ to

$$\pi_i(\mathbf{w}) := \int_{x=0}^{x=w_i} \xi_i(w_1, \ldots, w_{i-1}, x, w_{i+1}, \ldots, w_d) dx$$

and observe that $\frac{\partial \pi_i}{\partial w_i}(\mathbf{w}) = \xi_i(\mathbf{w})$ by the fundamental theorem of calculus. $\qquad\square$

**Proof of proposition 2.**  Before proving proposition 2, we first prove a lemma.

**Lemma 7** (generalized Helmholtz decomposition). *The Jacobian decomposes into $\mathbf{J}(\mathbf{w}) = \mathbf{S}(\mathbf{w}) + \mathbf{A}(\mathbf{w})$ where $\mathbf{S}(\mathbf{w})$ and $\mathbf{A}(\mathbf{w})$ are symmetric and antisymmetric, respectively, for all $\mathbf{w} \in \mathbb{R}^d$.*

*Proof.* Follows immediately. See Letcher et al. (2019) for details and explanation. $\qquad\square$

**Proposition 2.** *Sentiment is additive: $D_{\mathbf{v}} \mathfrak{f}_{\mathbf{v}}(\mathbf{w}) = \sum_i D_{\mathbf{v}_i} \mathfrak{f}_{\mathbf{v}_i}(\mathbf{w})$.*

*Proof.* For any collection of updates $(\mathbf{v}_i)_{i=1}^n$, we need to show that

$$\mathbf{v}^\mathsf{T} \cdot \nabla \mathfrak{f}_{\mathbf{v}}(\mathbf{w}) = \sum_i \mathbf{v}_i^\mathsf{T} \cdot \nabla \mathfrak{f}_{\mathbf{v}_i}(\mathbf{w}).$$

Direct computation obtains $\mathbf{v}^\mathsf{T} \nabla \mathfrak{f}_{\mathbf{v}}(\mathbf{w}) = \mathbf{v}^\mathsf{T} \mathbf{J} \mathbf{v} = \mathbf{v}^\mathsf{T} \mathbf{S} \mathbf{v} + \mathbf{v}^\mathsf{T} \mathbf{A} \mathbf{v} = \sum_i \mathbf{v}^\mathsf{T} \mathbf{S}_{ii} \mathbf{v} = \sum_i \mathbf{v}_i^\mathsf{T} \cdot \nabla \mathfrak{f}_{\mathbf{v}_i}(\mathbf{w})$ because $\mathbf{A}$ is antisymmetric and $\mathbf{S}$ is block-diagonal. $\qquad\square$

**Proof of proposition 3.**    First we prove a lemma.

**Lemma 8.**  $\boldsymbol{\xi}_{\boldsymbol{\eta}}^{\mathsf{T}}(\mathbf{w}) \cdot \nabla \mathfrak{f}_{\boldsymbol{\eta}}(\mathbf{w}) = \sum_{i=1}^{n} \eta_i^2 \cdot \boldsymbol{\xi}_i^{\mathsf{T}} \nabla \mathfrak{f}_i(\mathbf{w}).$

*Proof.*  Observe by direct computation that

$$
\nabla \mathfrak{f}_i = \mathbf{J}^{\mathsf{T}} \cdot \begin{pmatrix} 0 \\ \vdots \\ \boldsymbol{\xi}_i \\ \vdots \\ 0 \end{pmatrix}, \quad \text{and so} \quad \nabla(\eta_i \mathfrak{f}_i) = \mathbf{J}^{\mathsf{T}} \cdot \begin{pmatrix} 0 \\ \vdots \\ \eta_i \cdot \boldsymbol{\xi}_i \\ \vdots \\ 0 \end{pmatrix}.
$$

It is then easy to see that $\nabla \mathfrak{f}_{\boldsymbol{\eta}} = \sum_i \eta_i \cdot \nabla \mathfrak{f}_i = \sum_i \eta_i \cdot \mathbf{J}^{\mathsf{T}} \boldsymbol{\xi}_i = \mathbf{J}^{\mathsf{T}} \boldsymbol{\xi}_{\boldsymbol{\eta}}$. Thus,

$$
\boldsymbol{\xi}_{\boldsymbol{\eta}}^{\mathsf{T}} \cdot \nabla \mathfrak{f}_{\boldsymbol{\eta}} = \boldsymbol{\xi}_{\boldsymbol{\eta}}^{\mathsf{T}} \cdot \mathbf{J}^{\mathsf{T}} \cdot \boldsymbol{\xi}_{\boldsymbol{\eta}} = \boldsymbol{\xi}_{\boldsymbol{\eta}}^{\mathsf{T}} \cdot (\mathbf{S} + \mathbf{A}^{\mathsf{T}}) \cdot \boldsymbol{\xi}_{\boldsymbol{\eta}}
$$

where $\mathbf{S} = \mathbf{S}^{\mathsf{T}}$ since $\mathbf{S}$ is symmetric. By antisymmetry of $\mathbf{A}$, we have that $\mathbf{v}^{\mathsf{T}} \mathbf{A}^{\mathsf{T}} \mathbf{v} = 0$ for all $\mathbf{v}$. The expression thus simplifies to

$$
\boldsymbol{\xi}_{\boldsymbol{\eta}}^{\mathsf{T}} \cdot \nabla \mathfrak{f}_{\boldsymbol{\eta}} = \boldsymbol{\xi}_{\boldsymbol{\eta}}^{\mathsf{T}} \cdot \mathbf{S} \cdot \boldsymbol{\xi}_{\boldsymbol{\eta}} = \sum_{i=1}^{n} \eta_i^2 \cdot (\boldsymbol{\xi}_i^{\mathsf{T}} \cdot \mathbf{S}_{ii} \cdot \boldsymbol{\xi}_i) = \sum_{i=1}^{n} \eta_i^2 \cdot \boldsymbol{\xi}_i^{\mathsf{T}} \nabla_i \mathfrak{f}_i
$$

by the block-diagonal structure of $\mathbf{S}$.  $\square$

**Proposition 3** (legibility under gradient dynamics). *Fix dynamics $\frac{d\mathbf{w}}{dt} := \boldsymbol{\xi}_{\boldsymbol{\eta}}(\mathbf{w})$. Sentiment decomposes additively:*

$$
\frac{d\mathfrak{f}_{\boldsymbol{\eta}}}{dt} = \sum_i \eta_i \cdot \frac{d\mathfrak{f}_i}{dt}
$$

*Proof.*  Applying the chain rule obtains that

$$
\frac{d\mathfrak{f}_{\boldsymbol{\eta}}}{dt} = \left\langle \nabla_{\mathbf{w}} \mathfrak{f}_{\boldsymbol{\eta}}, \frac{d\mathbf{w}}{dt} \right\rangle = \left\langle \nabla_{\mathbf{w}} \mathfrak{f}_{\boldsymbol{\eta}}, \boldsymbol{\xi}_{\boldsymbol{\eta}}(\mathbf{w}) \right\rangle,
$$

where the second equality follows by construction of the dynamical system as $\frac{d\mathbf{w}}{dt} = \boldsymbol{\xi}_{\boldsymbol{\eta}}(\mathbf{w})$. Lemma 8 shows that

$$
\begin{aligned}
\left\langle \nabla_{\mathbf{w}} \mathfrak{f}_{\boldsymbol{\eta}}, \boldsymbol{\xi}_{\boldsymbol{\eta}}(\mathbf{w}) \right\rangle &= \boldsymbol{\xi}_{\boldsymbol{\eta}}^{\mathsf{T}} \mathbf{J}^{\mathsf{T}} \boldsymbol{\xi}_{\boldsymbol{\eta}} \\
&= \boldsymbol{\xi}_{\boldsymbol{\eta}}^{\mathsf{T}} (\mathbf{S} + \mathbf{A}^{\mathsf{T}}) \boldsymbol{\xi}_{\boldsymbol{\eta}} \\
&= \sum_i \eta_i^2 \cdot \boldsymbol{\xi}_i^{\mathsf{T}} \mathbf{S}_{ii} \boldsymbol{\xi}_i \\
&= \sum_i \eta_i^2 \left\langle \boldsymbol{\xi}_i, \nabla \mathfrak{f}_i(\mathbf{w}) \right\rangle.
\end{aligned}
$$

Finally, since $\frac{d\mathbf{w}_i}{dt} = \eta_i \cdot \boldsymbol{\xi}_i(\mathbf{w})$ by construction, we have

$$
\begin{aligned}
\left\langle \eta_i \cdot \boldsymbol{\xi}_i, \eta_i \cdot \nabla \mathfrak{f}_i(\mathbf{w}) \right\rangle &= \left\langle \frac{d\mathbf{w}_i}{dt}, \nabla_{\mathbf{w}_i} \left( \eta_i \mathfrak{f}_i(\mathbf{w}) \right) \right\rangle \\
&= \frac{d(\eta_i \mathfrak{f}_i)}{dt} = \eta_i \frac{d\mathfrak{f}_i}{dt}
\end{aligned}
$$

for all $i$ as required.  $\square$

**Proof of theorem 4.**

**Theorem 4.** *A fixed point in an SM-game is a local Nash equilibrium iff it is stable.*

*Proof.* Suppose that $\mathbf{w}^*$ is a fixed point of the game, that is suppose $\boldsymbol{\xi}(\mathbf{w}^*) = \mathbf{0}$.

Recall from lemma 7 that the Jacobian of $\boldsymbol{\xi}$ decomposes uniquely into two components $\mathbf{J}(\mathbf{w}) = \mathbf{S}(\mathbf{w}) + \mathbf{A}(\mathbf{w})$ where $\mathbf{S} \equiv \mathbf{S}^\mathsf{T}$ is symmetric and $\mathbf{A} + \mathbf{A}^\mathsf{T} \equiv 0$ is antisymmetric. It follows that $\mathbf{v}^\mathsf{T}\mathbf{J}\mathbf{v} = \mathbf{v}^\mathsf{T}\mathbf{S}\mathbf{v} + \mathbf{v}^\mathsf{T}\mathbf{A}\mathbf{v} = \mathbf{v}^\mathsf{T}\mathbf{S}\mathbf{v}$ since $\mathbf{A}$ is antisymmetric. Thus, $\mathbf{w}^*$ is a stable fixed point iff $\mathbf{S}(\mathbf{w}^*) \succ \mathbf{0}$ is negative definite.

In an SM-game, the antisymmetric component is arbitrary and the symmetric component is block diagonal – where blocks correspond to players' parameters. That is, $\mathbf{S}_{ij} = \mathbf{0}$ for $i \neq j$ because the interactions between players $i$ and $j$ are pairwise zero-sum – and are therefore necessarily confined to the antisymmetric component of the Jacobian. Since $\mathbf{S}$ is block-diagonal, it follows that $\mathbf{S}$ is negative definite iff the submatrices $\mathbf{S}_{ii}$ along the diagonal are negative definite for all players $i$.

Finally, $\mathbf{S}_{ii}(\mathbf{w}^*) = \nabla^2_{ii}\pi_i(\mathbf{w}^*)$ is negative definite iff profit $\pi_i(\mathbf{w}^*)$ is strictly concave in the parameters controlled by player $i$ at $\mathbf{w}^*$. The result follows. $\square$

## Proof of theorem 5.

**Theorem 5.** *In continuous time, for all positive learning rates $\boldsymbol{\eta} \succ \mathbf{0}$,*

1. *If $\mathbf{w}^*$ is a stable fixed point ($\mathbf{S} \prec \mathbf{0}$), then there is an open neighborhood $U \ni \mathbf{w}^*$ where $\frac{d\mathfrak{f}_{\boldsymbol{\eta}}}{dt}(\mathbf{w}) < 0$ for all $\mathbf{w} \in U \setminus \{\mathbf{w}^*\}$, so the dynamics converge to $\mathbf{w}^*$ from anywhere in $U$.*

2. *If $\mathbf{w}^*$ is an unstable fixed point ($\mathbf{S} \succ \mathbf{0}$), there is an open neighborhood $U \ni \mathbf{w}^*$ such that $\frac{d\mathfrak{f}_{\boldsymbol{\eta}}}{dt}(\mathbf{w}) > 0$ for all $\mathbf{w} \in U \setminus \{\mathbf{w}^*\}$, so the dynamics within $U$ are repelled by $\mathbf{w}^*$.*

*Proof.* We prove the first part. The second follows by a symmetric argument. First, strict concavity implies $\mathbf{S}_{ii} = \nabla^2_{ii}\pi_i$ is negative definite for all $i$. Second, since $\mathbf{S}$ is block-diagonal, with zeros in all blocks $\mathbf{S}_{ij}$ for pairs of players $i \neq j$, it follows that $\mathbf{S}$ is also negative definite. Observe that

$$\boldsymbol{\xi}_{\boldsymbol{\eta}}^\mathsf{T} \cdot \nabla\mathfrak{f}_{\boldsymbol{\eta}} = \boldsymbol{\xi}_{\boldsymbol{\eta}}^\mathsf{T} \cdot \mathbf{J}^\mathsf{T}\boldsymbol{\xi}_{\boldsymbol{\eta}} = \boldsymbol{\xi}_{\boldsymbol{\eta}}^\mathsf{T} \cdot (\mathbf{S} + \mathbf{A}^\mathsf{T})\boldsymbol{\xi}_{\boldsymbol{\eta}} = \boldsymbol{\xi}_{\boldsymbol{\eta}}^\mathsf{T} \cdot \mathbf{S} \cdot \boldsymbol{\xi}_{\boldsymbol{\eta}} < 0$$

for all $\boldsymbol{\xi}_{\boldsymbol{\eta}} \neq \mathbf{0}$ since $\mathbf{S}$ is negative definite. Thus, simultaneous gradient ascent on the profits acts to infinitesimally reduce the function $\mathfrak{f}_{\boldsymbol{\eta}}(\mathbf{w})$.

Since $\boldsymbol{\xi}_{\boldsymbol{\eta}}$ reduces $\mathfrak{f}_{\boldsymbol{\eta}}$, it will converge to a stationary point satisfying $\nabla\mathfrak{f}_{\boldsymbol{\eta}} = \mathbf{0}$. Observe that $\nabla\mathfrak{f}_{\boldsymbol{\eta}} = \mathbf{0}$ iff $\boldsymbol{\xi}_{\boldsymbol{\eta}} = \mathbf{0}$ since $\nabla\mathfrak{f}_{\boldsymbol{\eta}} = \mathbf{J}^\mathsf{T}\boldsymbol{\xi}_{\boldsymbol{\eta}}$ and the symmetric component $\mathbf{S}$ of the Jacobian is negative definite. Finally, observe that all stationary points of $\mathfrak{f}_{\boldsymbol{\eta}}$, and hence $\boldsymbol{\xi}_{\boldsymbol{\eta}}$, are stable fixed points of $\boldsymbol{\xi}_{\boldsymbol{\eta}}$ because $\mathbf{S}$ is negative definite, which implies that the fixed point is a Nash equilibrium. $\square$

## Proof of theorem 6.

**Theorem 6.** *Suppose all firms have negative sentiment, $\frac{d\mathfrak{f}_i}{dt}(\mathbf{w}) < 0$, for sufficiently large values of $\|\mathbf{w}_i\|$. Then the dynamics are bounded for any learning rates $\boldsymbol{\eta} \succ \mathbf{0}$.*

*Proof.* Fix $\boldsymbol{\eta} \succ \mathbf{0}$ and also fix $d > 0$ such that $\frac{d\mathfrak{f}_i}{dt}(\mathbf{w}) < 0$ for all $\mathbf{w}$ satisfying $\|\mathbf{w}_i\|_2 > d$. Let $U(d) = \{\mathbf{w} : \|\mathbf{w}_i\|_2 > d \text{ for all } i\}$ and suppose $g > 0$ is sufficiently large such that $\mathfrak{f}_{\boldsymbol{\eta}}^{-1}(g) = \{\mathbf{w} : \mathfrak{f}_{\boldsymbol{\eta}}(\mathbf{w}) = g\} \subset U(d)$. We show that

$$\mathfrak{f}_{\boldsymbol{\eta}}(\mathbf{w}^{(0)}) < g \quad \text{implies} \quad \mathfrak{f}_{\boldsymbol{\eta}}(\mathbf{w}^{(t)}) < g \quad \text{for all } t > 0,$$

for the dynamical system defined by $\frac{d\mathbf{w}}{dt} = \boldsymbol{\xi}_{\boldsymbol{\eta}}$. Since we are operating in continuous time, all that is required is to show that $\mathfrak{f}_{\boldsymbol{\eta}}(\mathbf{w}^{(t)}) = g' < g$ implies that $\mathfrak{f}_{\boldsymbol{\eta}}(\mathbf{w}^{(t+\epsilon)}) < g'$ for all sufficiently small $\epsilon > 0$.

Recall that $\frac{d\mathfrak{f}_i}{dt}(\mathbf{w})(\mathbf{w}) := \boldsymbol{\xi}_i^\mathsf{T} \cdot \nabla^2_{ii}\pi_i \cdot \boldsymbol{\xi}_i = D_{\boldsymbol{\xi}_i}(\frac{1}{2}\|\boldsymbol{\xi}_i\|_2^2)$. It follows immediately that $D_{\boldsymbol{\xi}_{\boldsymbol{\eta}}}(\frac{1}{2}\|\boldsymbol{\xi}_{\boldsymbol{\eta}}\|_2^2) = \sum_i \eta_i \cdot \frac{d\mathfrak{f}_i}{dt}(\mathbf{w}) < 0$ for all $\mathbf{w}$ in a sufficiently small ball centered at $\mathbf{w}^{(t)}$. In other words, the dynamics $\frac{d\mathbf{w}}{dt} = \boldsymbol{\xi}_{\boldsymbol{\eta}}$ reduce $\mathfrak{f}_{\boldsymbol{\eta}}$ and the result follows. $\square$

## C  Near SM-games: Experiential Value and the Exchange of Goods

Definition 3 proposes a model of *monetary* exchange in smooth markets. It ignores some major aspects of actual markets. For example, SM-games do not model inventories, investment, borrowing or interest rates. Moreover, in practice money is typically exchanged in return for goods or services – which are ignored by the model.

In this section, we sketch one way to extend SM-games to model the exchange of both money and goods - although still without accounting for inventories, which would more significantly complicate the model. The proposed extension is extremely simplistic. It is provided to indicate how the model's expressive power can be increased, and complications that results.

Suppose

$$\pi_i(\mathbf{w}) = f_i(\mathbf{w}) + \sum_{j \neq i} \Big( \alpha_{ij} \omega_{ij}(\mathbf{w}_i, \mathbf{w}_j) - g_{ij}(\mathbf{w}_i, \mathbf{w}_j) \Big).$$

The functions $\omega_{ij}$ measure the amount of goods (say, widgets) that are exchanged between firms $i$ and $j$. We assume that $\omega_{ij} + \omega_{ji} \equiv 0$ since widgets are physically passed between the firms and therefore one firms increase must be the others decrease. For two firms to enter into an exchange it must be that they subjectively value the widgets differently, hence we introduce the parameters $\alpha_{ij}$. Note that if $\alpha_{ij} = 1$ for all $ij$ then the model is equivalent to an SM-game.

The transaction between firms $i$ and $j$ is net beneficial to *both* firms $i$ and $j$ if

$$\alpha_{ij} \cdot \omega_{ij}(\mathbf{w}_i, \mathbf{w}_j) > g_{ij}(\mathbf{w}_i, \mathbf{w}_j)$$

and, simultaneously

$$\alpha_{ji} \cdot \omega_{ji}(\mathbf{w}_i, \mathbf{w}_j) > g_{ji}(\mathbf{w}_i, \mathbf{w}_j).$$

We can interpret the inequalities as follows. First suppose that $\omega_{ij}$ and $g_{ij}$ always have the same sign. The assumption is reasonable so long as firms do not pay to *give away* widgets. Further assume without loss of generality that $\omega_{ij}$ and $g_{ij}$ are both greater than zero – in other words, firm $i$ is buying widgets from firm $j$. The above inequalities can then be rewritten as

$$\underbrace{\alpha_{ij} \cdot \omega_{ij}(\mathbf{w}_i, \mathbf{w}_j)}_{\text{amount firm } i \text{ values the widgets}} > \underbrace{g_{ij}(\mathbf{w}_i, \mathbf{w}_j)}_{\text{amount firm } i \text{ pays}}$$

and

$$\underbrace{\alpha_{ji} \cdot \omega_{ij}(\mathbf{w}_i, \mathbf{w}_j)}_{\text{amount } j \text{ values the widgets}} < \underbrace{g_{ij}(\mathbf{w}_i, \mathbf{w}_j)}_{\text{amount } j \text{ is paid}}$$

It follows that both firms benefit from the transaction.

**Implications for dynamics.** The off-block-diagonal terms of the symmetric and anti-symmetric components of the game Jacobian are

$$\mathbf{S}_{ij} = \frac{\alpha_{ij} - \alpha_{ji}}{2} \cdot \nabla^2_{ij} \omega_{ij}(\mathbf{w}_i, \mathbf{w}_j)$$

and

$$\mathbf{A}_{ij} = \frac{\alpha_{ij} + \alpha_{ji}}{2} \cdot \nabla^2_{ij} \omega_{ij}(\mathbf{w}_i, \mathbf{w}_j)$$

where it is easy to check that $\mathbf{S}_{ij} = \mathbf{S}_{ji}$ and $\mathbf{A}_{ij} + \mathbf{A}_{ji} = 0$. The off-block-diagonal terms of $\mathbf{S}$ has consequences for how forecasts behave:

$$\boldsymbol{\xi}_{\boldsymbol{\eta}}^{\mathsf{T}} \cdot \nabla \mathfrak{f}_{\boldsymbol{\eta}} = \sum_i \eta_i^2 \cdot \underbrace{(\boldsymbol{\xi}_i^{\mathsf{T}} \cdot \nabla^2_{ii} \pi_i \cdot \boldsymbol{\xi}_i)}_{\text{sentiment}} \tag{5}$$

$$+ \sum_{i \neq j} \eta_i \eta_j \cdot (\alpha_{ij} - \alpha_{ji}) \cdot \underbrace{(\boldsymbol{\xi}_i^{\mathsf{T}} \cdot \nabla_{ij} \omega_{ij} \cdot \boldsymbol{\xi}_j)}_{\text{correction terms}}$$

**When are near SM-games well-behaved?** If $\alpha_{ij} = \alpha_{ji}$ for all $i, j$ then the correction is zero; if $\alpha_{ij} \sim \alpha_{ji}$ then the corrections due to different valuations of goods will be negligible, and the game should be correspondingly well-behaved.

**What can go wrong?** Eq (5) implies that the dynamics of near SM-games – specifically whether the dynamics are increasing or decreasing the aggregate forecast – cannot be explained in terms of the sum of sentiments of individual terms. The correction terms involve interactions between dynamics of different firms and the (second-order) quantities of goods exchanged. In principle, these terms could be arbitrarily large positive or negative numbers.

Concretely, the correction terms involving couplings between dynamics of different firms can lead to positive feedback loops, as in example 1, where the dynamics spiral off to infinity even though both players have strongly concave profit functions.

## D    THE `STOP_GRADIENT` OPERATOR

**Lemma 9.** *Any smooth vector field can be constructed as the gradient of a function augmented with* `stop_gradient` *operators.*

*Proof.* Suppose $\boldsymbol{\xi} = (\frac{\partial f_1(\mathbf{w})}{\partial w_1}, \ldots, \frac{\partial f_d(\mathbf{w})}{\partial w_d})$. Define

$$g(\mathbf{w}) = \sum_{i=1}^{d} f\Big(w_i, \texttt{stop\_gradient}(\mathbf{w}_{\hat{i}})\Big)$$

It follows that

$$\nabla_{\text{AD}}\Big[g(\mathbf{w})\Big] = \boldsymbol{\xi}(\mathbf{w})$$

as required.                                                                    $\square$

