# OpenReview forum: "Smooth markets: A basic mechanism for organizing gradient-based learners"
_ICLR.cc/2020/Conference — Accept (Poster)_

### Official Review · AnonReviewer2 · 2019-10-20
**Official Blind Review #2**

**Rating:** 8

**Review:**

This paper introduces smooth market games (SM-games), a class of smooth games characterized by pairwise zero sum interactions, and show SM-games possess a number of appealing properties:
- A fixed point is a local Nash equilibrium iff it is stable
- Local convergence to stable fixed points
- Unstable fixed points are repellent
- Dynamics are bounded, assuming diminishing returns-to-scale for sufficiently large vectors.
The need for a class of games with such properties is motivated by a discussion of the pathologies of smooth games under simultaneous gradient ascent.

The paper has an extensive literature review, addresses an important problem, and has informative discussions. It is well written and there are nice examples to illustrate the central ideas. However, I do have some criticisms:

1. The writing in this paper is sometimes pompous. Quoting James Scott has no added value to this work. Reference to Adam Smith’s invisible hand not only has no added value but is also confusing and detracts from the paper — SM-games are not purporting to be real economic models.
2. Throughout the paper there is a confusing mix between ideas from economics and ideas from accounting. For example, it is not correct to say that aggregate revenue is the same as GDP. Revenue is an accounting term to show how much benefit you record in a year. GDP measures the value of production. It would be more accurate to write that aggregate output and GDP are the same. But if SM-games are reflecting the perspective of an accountant, as is stated, why is GDP being discussed at all? Similarly, the idea of a dummy player with off the book costs is not consistent with an accounting perspective. I think that the paper would be clearest if it did not attempt to use terminology from other fields. But if it is going to do so, it should make a more substantial effort to do so consistently.

**Experience Assessment:**

I have read many papers in this area.

**Review Assessment: Checking Correctness Of Derivations And Theory:**

I assessed the sensibility of the derivations and theory.

**Review Assessment: Checking Correctness Of Experiments:**

N/A

**Review Assessment: Thoroughness In Paper Reading:**

I read the paper thoroughly.

---

> ### Author Response · Authors · 2019-11-14
> **Author response**
>
> We thank the reviewer for their time and feedback. We have two responses:
>
> 1. Pompous writing.
>
> We agree with R2 and will tone down the writing -- see the next point.
>
> On the other hand, it’s worth defending the James Scott quote. As we see it, overly focusing on (Nash) equilibria has been a major blocker for research on n-player games. We find the Scott quote inspiring because it suggests to let go of equilibria and instead search for measurements that can be pieced together to provide a synoptic view (a map) of a game. That is, we don’t need to find Nash equilibria to understand what is happening or predict what will happen in an interacting population.
>
> 2. R2 points out that we used a confusing mix of terminology. We agree. In particular, the analogy with GDP adds nothing to the paper. We will edit the paper to make the terminology more clear and consistent.

---

### Official Review · AnonReviewer3 · 2019-10-24
**Official Blind Review #3**

**Rating:** 8

**Review:**

This paper draws from concepts and patterns of game theory and economics to re-interpret common machine learning algorithms, and introduces a paradigm of n-player games such that there exists a simple way to study such games where each player is individually optimized by machine learning algorithms.

Overall, this is a very interesting and inspiring paper with its interdisciplinary touch, but at the same time doesn’t lose readability for audience with mostly machine learning background. The paper is clear, concise, and well-written, and hence I do not have any overarching comments. There are several suggestions and questions, though, that I’d like to propose.

1. Since many concepts are not from the machine learning domain, further and more detailed touch on related work are very beneficial. The current section of related work (Section 1.2) is more succinct than laying the right background information. Authors could consider giving some examples of “market mechanism”, and similarly for “design pattern”. This can give readers some ideas on how this present work’s use of “existing design pattern” is different from prior works’ proposals of “market mechanism”. Again for the second paragraph, although these concepts are more familiar to the audience, a short description for each mentioned concept is good to include (e.g. monotone games, etc)

2. Move Lemma 1 before Definition 2? Since Lemma 1 is most related to Definition 1 and not related to Definition 2.

3. For Definition 2, authors could consider adding “the classic definition [of Nash equilibrium]” to the list to clarify the difference. This classic Nash equilibrium definition will be referred to again in later sections.

4. In Section 2.1, consider giving a formal definition of “potential game”.

5. In Section 2.3, it is not immediately clear after the text why stop_gradient becomes a problem.

6. In Section 3.1, could we elucidate a bit on what is “near zero sum”?

7. In Section 5, why the equation following the text “firm i’s forecast” is always positive? Since by definition, forecast is meant to represent the change of profit, which can be negative?


**Experience Assessment:**

I do not know much about this area.

**Review Assessment: Checking Correctness Of Derivations And Theory:**

I assessed the sensibility of the derivations and theory.

**Review Assessment: Checking Correctness Of Experiments:**

N/A

**Review Assessment: Thoroughness In Paper Reading:**

I read the paper thoroughly.

---

> ### Author Response · Authors · 2019-11-14
> **Author response**
>
> We thank the reviewer for their time and detailed feedback.
>
> 1. More background material.
> The reviewer is correct, the paper covers a wide range of topics quite rapidly. We will provide more discussion in the related work section and also the Appendix to help orient readers.
>
> 2. Move Lemma 1 before Definition 2?
> Yes, will do.
>
> 3. Use “classic definition of Nash”.
> Yes
>
> 4. Define potential games.
> We will put this in the appendix to save space.
>
> 5. How can stop_grad cause problems?
>
> Stop_grad can be used to construct essentially any smooth game, see discussion in Appendix C. Stop_grad thus opens the door to a huge variety of pathological behaviors and intractable dynamics. We will provide a more detailed explanation in the final version.
>
> 6. Near zero-sum.
>
> Although GANs are adversarial, they are not always zero-sum games. A concrete example is discussed in sections 3.2.2 and 3.2.3 of Goodfellow’s tutorial (https://arxiv.org/abs/1701.00160). In short, it turns out that gradients tend to saturate in the original minmax setting, so Goodfellow introduced a “heuristic, non-saturating game”. By now there is a huge number of GANs in the literature, and it is difficult to find any mathematical property that is common to all of them. In particular, we do not have a precise definition of “near zero-sum”. Nevertheless, GANs do share adversarial dynamics as a unifying theme. Appendix B of the paper contains a brief discussion of some implications of loosening the pairwise zero-sum constraint.I
>
> 7. Why are forecasts always positive?
>
> Section 5 imposes the condition that “production updates” are either gradients or gradients rescaled by positive learning rates. Firms that update their production vectors using gradients will therefore always forecast profits to infinitesimally increase -- precisely because the updates are in the direction of steepest ascent. However, in reality, profits may of course go down due to the actions of other firms.

---

### Decision · Program_Chairs · 2019-12-19

**Decision:**

Accept (Poster)

**Comment:**

The paper discusses smooth market games and demonstrate the merit of the approach.    The reviewers agree on the quality of the paper, and the comments have been addressed well by the authors.